# Identification of a *bla*_VIM-1_-Carrying IncA/C_2_ Multiresistance Plasmid in an *Escherichia coli* Isolate Recovered from the German Food Chain

**DOI:** 10.3390/microorganisms9010029

**Published:** 2020-12-24

**Authors:** Natalie Pauly, Jens Andre Hammerl, Mirjam Grobbel, Annemarie Käsbohrer, Bernd-Alois Tenhagen, Burkhard Malorny, Stefan Schwarz, Diana Meemken, Alexandra Irrgang

**Affiliations:** 1Department Biological Safety, German Federal Institute for Risk Assessment, 10589 Berlin, Germany; Jens-Andre.Hammerl@bfr.bund.de (J.A.H.); mirjam.grobbel@bfr.bund.de (M.G.); annemarie.kaesbohrer@bfr.bund.de (A.K.); Bernd-Alois.Tenhagen@bfr.bund.de (B.-A.T.); Burkhard.Malorny@bfr.bund.de (B.M.); 2Institute for Veterinary Public Health, University of Veterinary Medicine, 1210 Vienna, Austria; 3Institute of Microbiology and Epizootics, Freie Universität Berlin, 14163 Berlin, Germany; stefan.schwarz@fu-berlin.de; 4Institute of Food Safety and Food Hygiene, Freie Universität Berlin, 14163 Berlin, Germany; Diana.Meemken@fu-berlin.de

**Keywords:** carbapenem, metallo-β-lactamase, IncA/C_2_, WGS, *bla*_VIM-1_

## Abstract

Within the German national monitoring of zoonotic agents, antimicrobial resistance determination also targets carbapenemase-producing (CP) *Escherichia coli* by selective isolation from food and livestock. In this monitoring in 2019, the CP *E. coli* 19-AB01133 was recovered from pork shoulder. The isolate was assigned to the phylogenetic group B1 and exhibited the multi-locus sequence-type ST5869. Molecular investigations, including whole genome sequencing, of 19-AB01133 revealed that the isolate carried the resistance genes *bla*_VIM-1_, *bla*_SHV-5_ and *bla*_CMY-13_ on a self-transmissible IncA/C_2_ plasmid. The plasmid was closely related to the previously described VIM-1-encoding plasmid S15FP06257_p from *E. coli* of pork origin in Belgium. Our results indicate an occasional spread of the *bla*_VIM-1_ gene in Enterobacteriaceae of the European pig population. Moreover, the *bla*_VIM-1_ located on an IncA/C_2_ plasmid supports the presumption of a new, probably human source of carbapenemase-producing Enterobacteriaceae (CPE) entering the livestock and food chain sector.

## 1. Introduction

Carbapenems are last resort antimicrobial agents against several infections caused by multidrug-resistant bacteria in human medicine. The World Health Organization defines them as “Highest Priority Critically Important Antimicrobials”. Nevertheless, the reports of carbapenem resistant bacteria are not limited to human medicine [1]. The repeated detection of carbapenemase-producing (CP) Enterobacteriaceae (CPE) in the German food-production chain is of considerable concern [1,2]. In general, CPE have been isolated sporadically from non-human sources (i.e., livestock, pets, wildlife) although carbapenems are not approved for veterinary application [2]. One of the main mechanisms of carbapenem resistance is the production of degrading enzymes, so called carbapenemases, which can hydrolyze almost all available β-lactams, including carbapenems [3]. The corresponding genes are often located on mobile genetic elements, in particular plasmids, which often carry additional antimicrobial, biocide, or heavy metal resistance genes [4]. These plasmids can be horizontally spread to bacteria under the selective pressure also imposed by non-β-lactam agents [5,6]. Since 2011, CPE isolates, which harbored VIM-1 encoding IncHI2 plasmids, were detected in livestock and food [7,8,9,10,11,12]. The first report described *Salmonella* Infantis and *E. coli* isolates detected in 2011 from fattening pigs in Germany [10]. They could persist in the animal host and closely related isolates could be detected in fattening pigs and minced meat in the following years [7,8]. Another VIM-1 report described the detection of this gene on an IncY plasmids in *E. coli* originated from seafood samples in Germany [13]. Besides the increasing reports of VIM-1 producing Enterobacteriaceae in the German food chain, the variants of carbapenemase genes increased as well. In 2019 within the annual resistance monitoring in the food chain in Germany, three CP *E. coli* were isolated. These include an OXA-48 producing *E. coli* [14] and a GES-5 producing *E. coli* [15], both isolated from pig feces, and another VIM-1 producing *E. coli* from a pork sample. This *bla*_VIM-1_ harboring *E. coli* isolate 19-AB01133 and its comprehensive molecular characteristics are described in this study. It shows great similarities to an *E. coli* isolate described by Garcia-Graells et al. (2020) [16] that was isolated from a pork sample in Belgium. Both isolates harbored almost identical IncA/C_2_ plasmids, which encode the *bla*_VIM-1_ gene. Furthermore, the genome data hint to a specific human entry source.

## 2. Materials and Methods

### 2.1. Isolate Origin and Antimicrobial Susceptibility Testing

The *E. coli* isolate 19-AB01133 was obtained in 2019 in the framework of the annual resistance monitoring in the food chain in Germany. The selective isolation of the CP *E. coli* was performed at a Federal State Laboratory, following Commission Implementing Decision (CID) 2013/652/EC and protocols provided by the European Union Reference Laboratory for Antimicrobial Resistance (EURL-AR) (https://www.eurl-ar.eu/protocols.aspx). The *E. coli* isolate 19-AB01133 was recovered from a shoulder meat sample of a pig raised in Germany. The pig was fattened and slaughtered in Brandenburg. The isolate was sent to the German Federal Institute for Risk Assessment for further confirmation and phenotypic and genotypic characterization. Antimicrobial susceptibility testing according to the EN ISO20776-1:2006 was conducted by broth microdilution using commercial plates (Thermo Fisher Scientific, Schwerte, Germany) with antimicrobial substances and concentrations defined by CID 2013/652/EC. The minimum inhibitory concentration (MIC) values were interpreted based on epidemiological cut-off values defined by the European Committee on Antimicrobial Susceptibility Testing (EUCAST) (www.eucast.org) and fixed in CID 2013/652/EU. 

### 2.2. Phenotypic and Genotypic Characterization 

Species confirmation of the isolate was conducted by MALDI-TOF Microflex LT/SH (Bruker Daltonics, Bremen, Germany) according to the manufacturer’s recommendations. Therefore, α-Cyano-4-hydroxycinnamic acid (HCCA, Bruker, MA, USA) was used as the matrix. Molecular determination of the carbapenem-resistance genes and initial phylogenetic typing were performed as previously described [14]. The *E. coli* 19-AB01133 was subjected to S1-nuclease pulsed-field gel electrophoresis (S1-PFGE) [17] (https://www.cdc.gov/pulsenet/pathogens/protocols.html) followed by Southern blot hybridization [18] against a digoxigenin-labelled *bla*_VIM-1_ probe while using DIG EasyHyb and DIG Wash and Block Buffer Set (Roche Diagnostics, Mannheim, Germany) according to the manufacturer’s recommendations. The transmissibility of the plasmid was investigated by in vitro filter-mating studies using *E. coli* K12 J53 as the recipient. Moreover, the plasmid pEC19-AB01133 was extracted by using the CosMCPrep Plasmid Purification Kit (Beckman Coulter, Krefeld, Germany) according to the manufacturer’s recommendations. The plasmids were transformed into highly competent *E. coli* DH10B (ElectroMAXTM DH10BTM Cells; InvitrogenTM, Thermo Fisher Scientific, Schwerte, Germany) by electroporation using disposable cuvettes with 1 mm gap and, 1.8 kV (E = 18 kV/cm) in a Bio-Rad MicroPulser (Bio-Rad Laboratories, Feldkirchen, Germany). Potential transconjugants and transformants were analyzed for their plasmid content and resistance phenotype. The *E. coli* isolate 19-AB01133 was subjected to short-read (MiSeq, Illumina, CA, USA) and long-read whole genome sequencing (MinIon; Oxford Nanopore, Oxford, UK) followed by a hybrid assembly of the obtained sequences using Unicycler v0.4.4 under default parameters [19]. Based on this assembly, the multi-locus sequence-type (MLST) as well as resistance and virulence genes were determined using online tools that were provided by the Danish Technical University (http://www.genomicepidemiology.org). The annotation was carried out by RAST2 provided by PATRIC (www.patricbrc.org). The assembly of the plasmid pEC19-AB01133 was deposited in GenBank (NCBI) under the accession number MT682138.

### 2.3. Farm Investigation

Three months after detecting the isolate 19-AB01133, the farm and the corresponding slaughterhouse were re-investigated. Therefore, composite fecal samples (*n* = 8) were taken from all pens. Moreover, ten environmental samples were taken in the farm. These samples include cobwebs, dust, water trough, wet residues and sock swabs. In the slaughterhouse, additional 17 samples were taken, including minced and sausage meat, fat, a smear of work surfaces and drains, curing water, hand washing water and more. These samples were analyzed according to the EURL method combined with a second enrichment step in lysogeny broth (LB) supplemented with 1 mg/L cefotaxime (CTX) and in LB supplemented with 0.125 mg/L meropenem (MEM). Afterwards, 10 µL of all enrichments (BPW, LB + CTX and LB + MEM) were streaked out on self-made selective agar (McConkey agar (McC) supplemented with 0.125 mg/L MEM and McC supplemented with 0.125 mg/L MEM and 1 mg/L CTX) and on chromID^®^ CARBA SMART (bioMérieux, Nürtingen, Germany). Plates were incubated for 16–18 h at 37 ± 2 °C. Up to ten colonies with different morphologies were picked and further analyzed by real-time PCR to confirm the presence or absence of *bla*_GES_, *bla*_KPC_, *bla*_NDM_, *bla*_OXA-48_ and *bla*_VIM_ [14] and by MALDI-ToF MS for species confirmation of presumptive CPE.

## 3. Results and Discussion

The phenotypic analysis of *E. coli* 19-AB01133 indicated a non-wildtype phenotype for the tested β-lactam antimicrobial agents, including penicillins (ampicillin MIC > 64 mg/L), cephalosporins (cefotaxime MIC 64 mg/L, ceftazidime MIC > 128 mg/L, cefepime MIC > 32 mg/L, cefoxitin MIC > 64 mg/L, ceftazidime MIC > 8 mg/L) and carbapenems (imipenem MIC 8 mg/L, ertapenem MIC > 2 mg/L and meropenem MIC 8 mg/L), and also for (fluoro)quinolones (ciprofloxacin MIC > 8 mg/L, nalidixic acid MIC > 128 mg/L) and aminoglycosides (gentamicin MIC > 32 mg/L) (Table 1). Further typing assigned the isolate to the phylogenetic group B1. This group is often associated with enhanced antimicrobial resistance but low virulence potential [20,21]. The virulence potential of the *E. coli* 19-AB01133 is composed of the genes *iss*, *ifpA* and *gad*, which code for an increased serum survival, for long polar fimbriae and for a glutamate decarboxylase, respectively. The S1-PFGE with subsequent Southern blot hybridization and in vitro filter-mating and transformation studies identified a single, conjugative ~190 kb plasmid carrying the *bla*_VIM-1_ gene. 

The bioinformatic analysis revealed assigned *E. coli* 19-AB01133 to the sequence type ST5869, which has not been described before from livestock and food in Germany. The plasmid pEC19-AB01133 (190,205 bp) exhibited an IncA/C_2_ replicon type. The characteristics of the presented isolate 19-AB01133 and its plasmid pEC19-AB01133 are summarized in the Appendix A. The encoded *bla*_VIM-1_ gene is highly likely to be responsible for the resistance to the tested carbapenems (imipenem, ertapenem and meropenem) in the *E. coli* isolate 19-AB01133. To the best of our knowledge, *bla*_VIM-1_-carrying IncA/C_2_ plasmids were detected sporadically in human samples [22,23], but so far only once in livestock and food in Europe [16]. Recently, Garcia-Grealls and her colleagues (2020) described an almost identical VIM-1 encoding plasmid (99.99% identical over the entire plasmid nucleotide sequence, Acc. No. PRJNA564835) of *E. coli* (Acc. No. MN477204.1) isolated from minced pork in Belgium in 2015 [16]. The corresponding isolate was assigned to the same sequence type, which might indicate a clonal relationship. This observation leads to the question of repeated contamination of pigs and pork with this strain, or to a persistence of this strain in Central European pig production. The *E. coli* isolate 19-AB01133 further showed a close relationship to the *E. coli* isolate TZ 116, which is used by the EURL-AR as second (backup) strain for the validation of selective plates for the detection of CP *E. coli* (https://www.eurl-ar.eu/protocols.aspx). This backup strain was isolated from a human sample. According to personal communication, the strain was not used in the isolating laboratory at that time or in our lab and thus, a cross-contamination can be excluded. 

Apart from this control strain, some more *bla*_VIM-1_ harboring IncA/C_2_ plasmids of isolates from human origin showed a close relationship to the described plasmid pEC19-AB01133. For example, Drieux et al. (2012) [23] described a plasmid (Acc. No. JQ824049) from a Greek *Providencia stuartii* multiresistant strain. By comparing this plasmid with the plasmid pEC19-AB01133, a query coverage of 87% was observed. Further, plasmids from Enterobacteriaceae of clinical and environmental samples from Canada showed also high similarities [24] to the described plasmid pEC17-AB01133. The plasmid pIncAC-KP4898 (Acc. No. KY882285.1) and the plasmid pRIVM0001_VIM-1 (Acc. No. MH220284.1) had a query coverage of 70% and the plasmid pKPC_CAV1344 (Acc. No. CP011622.1) a query coverage of 81%. Taken together, the close relationship and the repeated detection of the strain in food chain samples hints to a spill-over of the strain from humans to food. 

The plasmid pEC19-AB01133 harbored a variety of antimicrobial resistance genes, which are located within a ~60 kb multiresistance region (Figure 1). The *bla*_VIM-1_ gene is located on a class 1 integron. In addition, the displayed region exhibited three further interesting segments. The first segment comprised two mercury resistance operons (Figure 1), which confer narrow-spectrum resistance to inorganic mercury [25]. Two *tnpA* genes encompass the 5768 bp segment 2. This DNA-segment, originally identified in *Citrobacter freundii* (Acc. No. AY339625.2), encodes the small multidrug resistance efflux transporter SugE and the AmpC β-lactamase CMY-13 [26]. To date, the occurrence of *bla*_CMY-13_ has only sporadically been reported [26]. Another neighboring class 1 integron conferred resistance to aminoglycosides [*aac(6′)-Il, aac(3)-l, ant(3”)-Ia*] and to sulfonamides (*sul1*). Downstream of this class 1 integron, the third segment was integrated. This segment carried the ESBL β-lactamase gene *bla*_SHV-5_. The *bla*_SHV-5_ gene has been previously reported from isolates of *E. coli*, *Klebsiella pneumoniae* and *Providencia stuartii* from humans, livestock and wildlife [27], but not yet in animals or food in Germany. 

The farm and the corresponding slaughter facility were re-investigated three months later. Thereby, 35 samples from feces, farm and slaughter facility environment and different meat products were taken. No CPE were detected in these samples. This suggests that the strain was unable to persist on the farm and spread of the VIM-1 plasmid did not occur. Nevertheless, the detection of an isolate with such high similarities to the Belgian isolates might hint to the ability for it to persist within the pork production chain [15]. A re-introduction by human, i.e., by farmers or workers, is also possible as there was no obvious link between the two fattening pig farms.

The detection of multiresistance IncA/C_2_ plasmids identified in *E. coli* isolates from food chain is a threat to public health. Previous studies from Germany described the location of *bla*_VIM-1_ only on IncHI2 or IncY plasmids. However, for IncA/C_2_ plasmids, a high impact on the dissemination of antimicrobial resistance genes has been reported [20]. The detection of new plasmid types and the recurrence of carbapenemase-producing isolates in the German pork production chain emphasizes the importance of the CPE monitoring [4,8,9]. The occurrence of the *bla*_VIM-1_ located on an IncA/C_2_ plasmid in an *E. coli* isolate from pork suggests a new, probably human source of CPE entering the food chain. 

## Figures and Tables

**Figure 1 microorganisms-09-00029-f001:**
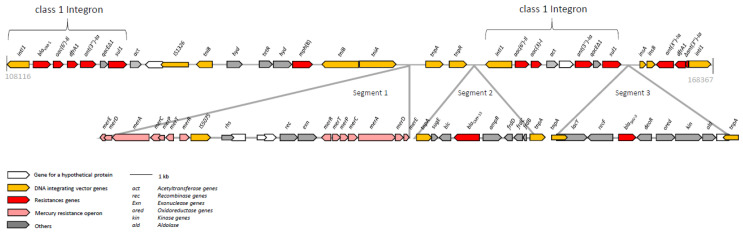
Schematic illustration of the multidrug-resistance region of plasmid pEC19-AB01133. The reading frames are displayed as arrows with the arrowhead showing the direction of transcription. The numbers refer to the whole plasmid sequence of pEC19-AB01133, which is deposited in the GenBank database under accession no. MT682138.

**Table 1 microorganisms-09-00029-t001:** Minimal inhibitory concentrations (MIC) of following antimicrobial agents (mg/L) to 19-AB01133 and its transkonjugant TK_19AB01133. MIC values classified as non-wildtype are colored in red. Ampicillin (AMP); Azithromycin (AZI); Cefepime (FEP); Chloramphenicol (CHL); Ciprofloxacin (CIP); Colistin (COL); Ertapenem (ERP); Cefotaxime (FOT); Cefoxitin (FOX); Gentamicin (GEN); Imipenem (IMI); Kanamycin (KAN); Meropenem (MERO); Nalidixic acid (NAL); Sulfamethoxazole (SMX); Ceftazidime (TAZ); Tetracycline (TET); Tigecycline (TGC); Trimethophrim (TMP).

Isolate	AMP	AZI	CHL	CIP	COL	ERP	FEP	FOT	FOX	GEN	IMI	MERO	NAL	SMX	TAZ	TET	TGC	TMP
19-AB01133	>64	16	16	>8	≤1	>2	>32	>64	>64	32	8	8	>128	>1024	>8	4	≤0.25	>32
TK_19-AB01133	>64	8	8	≤0.015	≤1	>2	>32	>64	>64	4	8	4	≤4	>1024	>8	4	≤0.25	>32
K12 J53	4	8	≤8	≤0.015	≤1	≤0.015	≤0.06	≤0.25	4	2	0.25	≤0.03	≤4	≤8	≤0.5	4	0.5	≤0.25

## Data Availability

Data is contained within the article and supplementary material. Moreover, the plasmid
sequence was deposited in GenBank (NCBI) under the accession number MT682138.

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
