# Peer review of "Identification of a blaVIM-1-Carrying IncA/C2 Multiresistance Plasmid in an Escherichia coli Isolate Recovered from the German Food Chain"

_microorganisms, 2020, doi:10.3390/microorganisms9010029_

Round 1

Reviewer 1 Report

The manuscript by Pauly et al. describes a single isolate of carbapenem resistant E.coli obtained from a pig meat sample in a slaugterhouse in Germany. The analysis and description of the isolate is performed in-depth. It would be beneficial to briefly describe a procedure used to obtain pEC19-AB01133 plasmid from the isolate. Otherwise the short Communivćation is written clearly and conveys the findings to the reader.

Author Response

We thank the reviewer for the positive feedback to our submitted manuscript. According to the suggestions of the reviewer, we had added the description of procedure for plasmid extraction (lines 81-83).

Reviewer 2 Report

In this study authors characterized a carbapenemase-producing (CP) Escherichia coli that was isolated from pork shoulder. Surveillance of antimicrobial resistance in food and livestock is highly important in controlling transmission of resistance.

Overall, this report is important, but this study is not sufficient to draw any strong conclusion.

Authors need to make sure that it is not a random contamination from human during the processing of meat. No further isolation of the resistant bug from the animal of the corresponding pig farm support a notion of random contamination event.

GenBank file is not accessible, please provide a link of the sequence. In addition, inclusion of a map of the plasmid in the manuscript is required.

Line 130: How identical these two isolates are?

How closely these isolates and plasmids are located in the phylogenetic tree compared to other similar isolates?

Line 133: Based on just 2 observations, authors should not claim repetitive contamination.

Line 177: Authors forgot to remove the comments of the internal review process.

Author Response

In this study authors characterized a carbapenemase-producing (CP) Escherichia coli that was isolated from pork shoulder. Surveillance of antimicrobial resistance in food and livestock is highly important in controlling transmission of resistance.

Overall, this report is important, but this study is not sufficient to draw any strong conclusion.

Response: We agree with the opinion of this reviewer, but the short communication was submitted for publication to report on the steadily increasing number of Carbapenemase-producing Enterobacteriaceae (esp. E. coli) in the German food production chain. Our study aims the occurrence of a blaVIM-1-carrying IncA/C2 plasmid, previously described in a Belgian pig and a human sample (TZ166) in the German food chain. The occurrence of this carbapenemase-encoding plasmid suggests a yet unknown link between the food chains of Belgium and Germany that need to be further determined. Beside many possibilities for the introduction of the plasmid-prototype to the German food chain (i.e., animal feeding, ……), we would also point out a potential human entry source.

Authors need to make sure that it is not a random contamination from human during the processing of meat. No further isolation of the resistant bug from the animal of the corresponding pig farm support a notion of random contamination event.

Response: A random contamination from human during the processing of meat can not be excluded. In fact, a human cross-contamination during slaughter or meat processing seems to be a possible explanation for the occurrence of the carbapenemase-encoding plasmid (see comment above).

GenBank file is not accessible, please provide a link of the sequence. In addition, inclusion of a map of the plasmid in the manuscript is required.

Response: The release of the sequence was initiated. According to the feedback of NCBI, the sequence should be available soon. We will further submit a copy of the plasmid sequence for evaluation by the reviewers. The multi-resistance region showing the genetic basis of the antimicrobial resistances (incl. the carbapenem resistance gene) is illustrated in figure 1. As the backbone of this plasmid is identical to the Belgium plasmid as referred in lines 130 to 133) the amount of novel information by adding a complete map of the plasmid would be no informative for the readers.

Line 130: How identical these two isolates are?

How closely these isolates and plasmids are located in the phylogenetic tree compared to other similar isolates?

Response: There are only plasmid sequences available from NCBI from the Belgium isolate. Both plasmids are 99.99 % identical over the entire plasmid nucleotide sequence (added in line 131 and 132). As no sequence data in the nucleotide database are available, a phylogenetic comparison of isolates is not possible. A potential a clonal relationship due to the same ST is addressed in line 133 and 134.

Line 133: Based on just 2 observations, authors should not claim repetitive contamination.

Response: According to the reviewers suggestion, we have changed the conclusion from “This observation suggests either repeated contamination of pigs and pork with this strain, or persistence of this strain in the central European pig production“ to “This observation leads to the question of repeated contamination of pigs and pork with this strain, or to a persistence of this strain in Central European pig production."  (Line 133, Line 134)

 Line 177: Authors forgot to remove the comments of the internal review process.

Response: We had deleted this comment in the revised version of the manuscript.

Round 2

Reviewer 2 Report

Thank you for providing the sequencing file, but a DNA sequence does not mean anything without having all the annotations.